# Tumor-Associated Macrophages in Gliomas—Basic Insights and Treatment Opportunities

**DOI:** 10.3390/cancers14051319

**Published:** 2022-03-04

**Authors:** Johannes K. Andersen, Hrvoje Miletic, Jubayer A. Hossain

**Affiliations:** 1Department of Biomedicine, University of Bergen, 5009 Bergen, Norway; johannes.andersen@uib.no (J.K.A.); hrvoje.miletic@uib.no (H.M.); 2Department of Pathology, Haukeland University Hospital, 5009 Bergen, Norway

**Keywords:** glioma, glioblastoma, immunotherapy, macrophage, microglia, tumor-associated macrophages (TAMs), targeted therapies, TAM biomarker, immune checkpoints

## Abstract

**Simple Summary:**

Macrophages are a specialized immune cell type found in both invertebrates and vertebrates. Versatile in functionality, macrophages carry out important tasks such as cleaning cellular debris in healthy tissues and mounting immune responses during infection. In many cancer types, macrophages make up a significant portion of tumor tissue, and these are aptly called tumor-associated macrophages. In gliomas, a group of primary brain tumors, these macrophages are found in very high frequency. Tumor-associated macrophages can promote glioma development and influence the outcome of various therapeutic regimens. At the same time, these cells provide various potential points of intervention for therapeutic approaches in glioma patients. The significance of tumor-associated macrophages in the glioma microenvironment and potential therapeutic targets are the focus of this review.

**Abstract:**

Glioma refers to a group of primary brain tumors which includes glioblastoma (GBM), astrocytoma and oligodendroglioma as major entities. Among these, GBM is the most frequent and most malignant one. The highly infiltrative nature of gliomas, and their intrinsic intra- and intertumoral heterogeneity, pose challenges towards developing effective treatments. The glioma microenvironment, in addition, is also thought to play a critical role during tumor development and treatment course. Unlike most other solid tumors, the glioma microenvironment is dominated by macrophages and microglia—collectively known as tumor-associated macrophages (TAMs). TAMs, like their homeostatic counterparts, are plastic in nature and can polarize to either pro-inflammatory or immunosuppressive states. Many lines of evidence suggest that immunosuppressive TAMs dominate the glioma microenvironment, which fosters tumor development, contributes to tumor aggressiveness and recurrence and, very importantly, impedes the therapeutic effect of various treatment regimens. However, through the development of new therapeutic strategies, TAMs can potentially be shifted towards a proinflammatory state which is of great therapeutic interest. In this review, we will discuss various aspects of TAMs in the context of glioma. The focus will be on the basic biology of TAMs in the central nervous system (CNS), potential biomarkers, critical evaluation of model systems for studying TAMs and finally, special attention will be given to the potential targeted therapeutic options that involve the TAM compartment in gliomas.

## 1. Introduction

GBM is one of the most common primary brain tumors. The incidence rate of GBM ranges from 0.59 to 5 per 100,000 [1], which increases dramatically after the age of 54 years [2]. Despite substantial research and increasing knowledge about the nature of this malignancy, a major breakthrough in treatment is still lacking. Key obstacles towards developing better treatment modalities include intra- and inter-tumoral heterogeneity and the presence of the blood–brain barrier (BBB) [3]. Today, even after debulking surgery followed by radiotherapy with the addition of temozolomide, the median survival is less than two years [4]. Based on molecular parameters, GBM is divided into a primary IDH-wildtype GBM, which develops de novo, and a secondary IDH-mutated GBM, which develops from lower-grade astrocytomas [5]. However, according to the recently published WHO classification, only the IDH-wildtype GBMs are designated as GBM, while the former IDH-mutated GBM lesions are now classified as IDH-mutated astrocytoma as they develop from lower grade IDH-mutated astrocytomas [6]. Both entities are diagnosed as WHO grade 4 tumors.

GBM interacts with a highly heterogeneous tumor microenvironment (TME). The TME consists of extracellular matrix components, soluble factors, different brain resident cells and various immune cell types in a unique combination that differs from normal brain tissue. Among the stromal cells, TAMs are the most abundant ones and are important contributors to disease progression [7,8]. Previous studies showed a positive correlation between the abundance of TAMs and glioma grade and a negative correlation with survival [9,10]. Immunotherapies such as immune checkpoint inhibitors and CAR T cells are effective for a subset of hematological malignancies but have not shown efficacy in GBM and other solid tumors [11]. A major reason for the lack of response to immunotherapy lies within the nature of GBM being an immunologically “cold” tumor, which implies that the TME is dominated by immunosuppressive TAMs and lacks substantial T cell infiltration. Thus, the immunosuppressive TME of GBM is an important obstacle for the development of future immunotherapies [12]. 

In this review, we will discuss the TAMs as a vital part of the GBM TME and important contributors to treatment resistance and progression. Finally, we will focus on the potential points of intervention to therapeutically target TAMs in GBM patients.

## 2. The Origin and Function of Macrophages and Microglia

Macrophages are evolutionarily conserved cell types that play important roles in host defense. While macrophages solely orchestrate immune response in primitive organisms, in higher organisms, macrophages and other myeloid cells interact closely with lymphocytes. Although macrophages are usually blood-derived and recruited in response to inflammatory signals, certain tissues host resident macrophages. For instance, the CNS contains a local macrophage—known as microglia. The origin of macrophages and microglia were debated for a long time, and it was thought that both cells develop from monocytes originating from a common hematopoietic stem cell that matures in the bone marrow [13]. It is now well-accepted that microglia, which represents around 10% of the brain cell population, are derived from the erythro-myeloid progenitors of the yolk sac during early embryonic development [14]. Under normal physiological conditions, the central nervous system is only occupied by resident microglia, and the presence of other bone-marrow-derived macrophages are associated with the diseased brain. Microglia are long-lived and have self-renewal capacity. As a result, they are not dependent on a constant influx to sustain their population [15,16].

Besides being an important part of the immune response, microglia play a critical role in the physiological homeostasis of the brain. In fact, microglia are important for brain development and function, such as neurogenesis and myelin formation [17]. Maintenance of brain homeostasis by microglia involves the removal of apoptotic cells and other debris with minimal or no immune stimulation. In contrast, macrophages infiltrate the brain only under pathological conditions to mediate phagocytosis of apoptotic cells or necrotic tissue fragments, which can lead to either an anti-inflammatory or pro-inflammatory reaction, respectively [18]. Macrophages and microglia detect apoptotic cells by sensing chemoattractants released by the dying cells. These so-called “find me” signals guide these cells toward the apoptotic cells and enable binding to the “eat me signals”. This cell-to-cell interaction initiates phagocytosis [19]. Phagocytosis of the apoptotic cells also forms the core of bridging the innate immune response with an adaptive immune response as macrophages and microglia, following phagocytosis, present relevant antigens to the T cells. Thus, it is not surprising that TAMs often exhibit dysfunctional antigen presentation capabilities [20].

## 3. Polarization of Microglia and Macrophages

Macrophages are historically known to exhibit high plasticity and, from a functional viewpoint, can be divided into pro-inflammatory (M1) and anti-inflammatory (M2) phenotypes, also known as classically activated and alternatively activated macrophages, respectively. Pro-inflammatory M1 macrophages are involved in anti-tumorigenic effects, while immunosuppressive M2 macrophages are associated with pro-tumorigenic mechanisms that promote tumor growth, angiogenesis and invasiveness [21]. This dichotomous way of TAM classification is indeed an analogy to the T-helper cells where Th1 is proinflammatory while Th2 is anti-inflammatory [22]. The existence of this inherent plasticity among the bone marrow-derived macrophages also seems to be a feature of microglia. Although the concept of microglial polarization is criticized [23], a number of studies show that microglia in the TME harbor polarization potential towards either M1- or M2-phenotypes [24,25,26,27,28,29].

However, it is now well-accepted that this M1/M2 dichotomy for TAMs is an oversimplification of more than two macrophage phenotypes [30,31]. In this regard, the M2 state is further subdivided into M2a, M2b, M2c and M2d states. These different states are based on their gene expression profile, effector function as well as chemokine and cytokine production profile [32]. A great deal of what we know about these distinct functional phenotypes is based on in vitro experiments using canonical chemokines that induce polarization. While TNF-α, IFN-γ and LPS drive polarization towards an M1 phenotype, IL-4, IL-10 and IL-13 induce an M2 shift [33]. In vivo experiments, however, show a more diverse and complex picture of this phenotypic reduction. In fact, single-cell RNA sequencing and immunohistochemical staining of tumor samples from both animals and humans uncovered that different polarization states coexist. Furthermore, single TAMs were analyzed, which revealed that both M1 and M2 signature can be detected at the same time within the same cell [34]. This supports the idea of plasticity and heterogeneous composition of TAMs in the TME and implies that the concept of M1/M2 is rather a continuum than a separate state of activation [35,36].

## 4. Markers for TAM

A distinct set of proteins is preferentially expressed in TAMs that helps distinguish different TAM-subsets from each other or from other immune cells. However, it is important to bear in mind that very few, if any, single markers can exclusively identify a given population or subset of TAMs. Thus, multiple markers are often used to identify certain TAM populations.

CD11b, also known as Integrin alpha M, is often used to identify TAMs. CD11b is a subunit of the heteromeric complement receptor Macrophage-1 antigen. CD11b is expressed on NK cell [37] and T cell subsets [38,39]. However, it is most abundantly expressed on monocytes and granulocytes. As a result, CD11b has become a widely used pan-myeloid marker. Considering the relatively low abundance of granulocytes and other subsets of CD11b^+^ lymphocytes compared to the TAMs in the TME [40,41], CD11b is sometimes used as a “TAM-marker” as well. CD68, ionized calcium-binding adaptor molecule 1 (Iba1), CD14 and F4/80 are more TAM-specific markers [42,43,44,45]. CD68, a glycoprotein mostly found in cytoplasmic granules, is a relatively reliable marker of murine macrophages. Although CD68 is used as a marker in humans too [46], caution should be applied because its expression in other types of human cells such as dendritic cells, endothelial cells and even some lymphoid subsets is also reported [47]. Iba1 is involved in the ruffling and phagocytosis of macrophage and microglia [48] and thus used as a marker for TAMs both in murine and human settings [44,49,50,51,52]. F4/80 is a murine TAM-specific marker and thus not applicable for TAMs of human origin. CD14 is widely used for human TAMs. It is noteworthy that CD14 is also found in monocytes. However, it is retained in tissue macrophages and also found in microglia [42,53] and thus can be used to identify human TAMs in various contexts.

To demarcate the two polarized modes of TAMs, various additional markers are proposed. In murine models of gliomas and other solid tumors, iNOS and Arginase 1 are reliably used as specific markers for M1 and M2 phenotypes, respectively, both in vitro and in vivo [54,55,56,57]. Indeed, classical Th1-cytokines such as IFN-γ and IL-1 can induce iNOS expression and Th2-cytokine such as IL-4 can upregulate Arginase 1 expression confirming the relative specificity of these markers [58,59]. Apart from iNOS and Arginase 1, a plethora of other markers are proposed for M1-M2 distinction, and among those, MHC-II, CD38, CD86, Gpr18, Fpr2, pSTAT1 for M1-like phenotype and CD206, Mgl2, Egr2, pSTAT3, pSTAT6 for M2-like phenotype are prominent [59,60,61,62,63]. A major problem in using markers for M1-M2 distinction is the incompatibility between mouse models and human tumors, including gliomas. Reliable markers for mice are not present in the human system and sometimes, if present, do not show association with polarization status [18]. For instance, the two most reliable mouse M1-M2 distinguishing markers, iNOS and Arginase 1, even though present in humans, do not comply with the TAM-polarization status [64]. Thus, more research is needed to identify more reliable markers for M1-M2 status in human TAMs.

Identifying reliable molecular markers differentiating infiltrating macrophages from resident microglia within the TAM population has also been a research focus for many years. Infiltrating macrophages, in general, express higher CD45 than microglia, and this phenomenon is used to differentiate these two cell types in flow-cytometric analyses [65]. However, in the TME, microglial cells can also acquire high CD45 expression undermining this strategy of phenotyping [66,67]. Recent studies have identified more specific markers such as the purinergic G protein-coupled receptor P2Y12 [45]. Other proposed markers for distinguishing microglial cells from the infiltrating macrophages are TMEM119 [68,69], Siglec-H [70,71], Olfml3 [72] and Sall1 [69,71]. In contrast, Integrin Subunit Alpha 4 or CD49D was reported to be specifically expressed on infiltrating macrophages and not on microglial cells [44,73]. Indeed, P2Y12 and CD49D were found to be expressed in a mutually exclusive manner in CNS TAMs, further supporting the specificity of these markers for microglia and macrophages, respectively [52].

## 5. Macrophages and Microglia in Glioma

The major immune cells in the glioma TME include microglia, monocyte-derived macrophages, neutrophils and T cells. Their distribution varies between different grades of glioma and metastatic brain tumors. In GBM, TAMs represent by far the most abundant cell type in the tumor mass and make up as much as 30–50% of the total mass [74]. Among the malignancies in the brain, IDH-wildtype GBM and metastatic brain tumors have the highest influx of monocyte-derived-macrophages [73]. Infiltration of TAMs tends to increase with tumor grade in different cancers, including GBM. Muller et al. found that the increase in infiltrating macrophages from the periphery was significantly higher in GBM compared to low-grade glioma. Furthermore, a higher infiltration of blood-derived TAMs was correlated with poor survival in low-grade gliomas, while no such correlation was observed between microglial density and survival [34].

TAMs in GBM demonstrate high expression of M2 polarization markers which is correlated with impaired prognosis and resistance to therapy [75]. Poon et al. found that IDH-mutant GBM had a lower proportion of TAMs compared to IDH-wildtype GBM, although there was a higher variation in the proportion of TAMs in IDH-wildtype compared to IDH-mutant GBM. The authors also observed that the TAMs in IDH-mutant GBM showed more pro-inflammatory characteristics compared to their wild-type counterparts. Interestingly, IDH-wildtype GBM had high intertumoral variability regarding pro- and anti-inflammatory TAMs [76]. However, TAM polarization in different glioma grades is diverse and not always clear. Immunohistochemical staining of different macrophage markers across gliomas showed a higher expression of the M2 marker CD163 in pilocytic astrocytoma (WHO grade I) compared to diffuse astrocytomas and GBM. Additionally, the expression of pan-macrophage markers was higher for pilocytic astrocytoma in the same study [36]. As pilocytic astrocytoma has a better prognosis compared to high-grade gliomas, the M1/M2 polarization scheme has a prognostic value only in certain tumors, but may not necessarily be correlated with high-grade gliomas. Several studies so far have shown an immunosuppressive TAM phenotype dominating the TME of gliomas. However, the exact phenotype of these TAMs in glioma is still highly controversial and ill-defined. Transcriptomic profiling of CD11b^+^ cells in GBM compared to healthy tissue did not demonstrate a clear shift towards either pro-tumorigenic or anti-tumorigenic polarization [31].

### 5.1. The Function of TAMs in the TME 

TAMs, in accordance with their plastic nature, can play a dual role during oncogenesis and treatment. In principle, TAMs have the potential to eradicate tumor cells, and not surprisingly, immunocompromised mice have poorer survival compared to immunocompetent mice when implanted with the same tumor cells [77]. Furthermore, TAM polarization is constantly shifting throughout tumor development and during tumor progression.

Interestingly, Maire et al. found that the highest expression of immune response-related genes in an immunocompetent GBM animal model was in the initial phase of tumor growth. Infiltration of TAMs, CD3^+^ and CD8^+^ T cells was augmented in the beginning and decreased after the mice became symptomatic. This illustrates the concept of immune editing and immune escape [78,79]. Maire et al. also observed that immunosuppressive cytokines, checkpoint ligands and surface molecules were expressed at much higher levels by the stromal cells, which mostly consisted of TAMs, compared to the tumor cells. Furthermore, the interferon regulatory factor 1 (IRF-1), a key transcriptional regulator, was activated in tumor cells through stimulation by cytokines released from TAMs. IRF-1-upregulated genes are linked to both immune escape (e.g., Programmed death-ligand 1 or PD-L1, CTLA4, SOCS1, Arginase 1) and immune stimulation (e.g., CXCL9, CXCL10, CIITA, CCL6). IRF-1 activation resulted in tumors tending to escape the immune system and less clonal diversity compared to tumors growing in immunocompromised mice [78]. This indicates that the immune system primes tumor cells for an immune escape and shapes the clonality of tumor cells towards a more immune evasive phenotype.

This concept of immune escape was also demonstrated in an experiment where tumors were reimplanted after being exposed to the TME. Mesenchymal GBM, a major subgroup defined by transcriptional signature, has an increased immune infiltration and has the poorest survival rate compared to the other signatures such as pro-neural and classical [80]. In a study with immunocompetent mice, a genetically engineered mesenchymal glioma cell line was re-implanted to obtain an immune evasive (IE) cell line. When comparing the TME in the original cell line with the reimplanted IE cell line in vivo, there was a significant increase in infiltration of monocytic myeloid-derived suppressor cells and macrophages. The macrophages showed higher expression of PD-L1 compared to microglia. Furthermore, the authors also found an upregulation of the chemokine CCL9 and the transcription factor IRF-8 in the more immune evasive cell line. The IRF-8 promoter region was shown to be hypomethylated consistent with epigenetic immunoediting. The authors also found loss of methylation in other immune evasive regulators such as Nt5e (CD73) and PD-L1 [77].

Resident microglia can also change their signatures on both transcriptional and protein levels from a homeostatic signature to an increased expression of pro-inflammatory and metabolic genes [81]. So far, microglia diversity of transcriptional states is much less explored compared to those of monocyte-derived macrophages in the glioma TME. Thus, more research is needed in this area, which is highlighted by recent studies showing wide variety in the transcriptional spectrum among microglia associated with pathological processes [82].

TAM-mediated immunosuppression in the glioma TME involves the lymphoid lineage cells as well. The role of T lymphocytes in this process is extensively studied. TAMs were found to obstruct the T cell-mediated pro-inflammatory response at various stages of T cell biology. Firstly, TAMs impede T cell infiltration into the GBM tissue. Interestingly, this inhibition of entry works selectively whereby the regulatory T cells (Tregs) are less affected than the Th1 cells [83]. This selective inhibition of T cell infiltration seems to be orchestrated by TAMs, which up- or downregulate various leukocyte chemoattractants. CCL2 (also known as MCP1, monocyte chemoattractant protein 1), a major monocyte chemoattractant that is highly expressed by TAMs, plays a key role in this process. CCL2 is a moderate-affinity ligand for the CCR4 receptor, which is the dominant-type chemoattractant receptor for Tregs and Th2 cells [84,85], whereas the Th1 cells prefer CXCR3 receptors [84]. As a result, TAM-derived CCL2 selectively attracts Tregs in the glioma microenvironment [85,86]. Epithelial membrane protein 3 (EMP3) was found to be involved in the process of CCL2-induction in GBM [86]. In hepatoma models, TAMs was shown to utilize the hedgehog pathway to suppress TAM-derived secretion of CXCL9 and CXCL10, which results in the reduction of migratory capacity of the CD8^+^ T cells, but not of CD4^+^ T cells [87]. CXCL9 and CXCL10 are also suppressed in TAMs of glioma by an EMP3-dependant process that reduces T cell infiltration [86]. In addition to blocking the entry, TAMs also sabotage the T cell functionality in glioma tissue. Indeed, the T cell population found in GBM are known to be severely dysfunctional. Various immunosuppressive molecules secreted by TAMs play critical roles in this process. For example, TGFβ, which is secreted by TAMs (and also GBM cells), can suppress the activity of cytotoxic T lymphocytes by reducing the key cytolytic products viz. perforin, granzyme A, granzyme B, Fas ligand and IFN-γ [88]. Another prominent molecule expressed by TAMs that can compromise T cell functionality is Arginase 1, which can curb T cell activation and proliferation by depleting the essential amino acid arginine [89,90]. Recently, the aryl hydrocarbon receptor (AHR) was also implicated in the process of TAM-directed modulation of T cell immunity [91]. AHR, abundantly expressed in high-grade gliomas, can compromise T cell immunity by reducing the expression of IFN-γ, TNF-α and granzyme B in T cells through adenosine production [91]. TAMs also promote T cell exhaustion in GBM by expressing various inhibitory molecules on their surface, which are collectively known as ‘immune checkpoints’ that trigger restrictive signaling in T cell proliferation and activation [83,92,93]. The role of immune checkpoints in T cell exhaustion is widely studied, and indeed, inhibitors of immune checkpoints (ICI) have emerged as a revolutionary class of anti-cancer drugs over the last years. Apart from modulating the functionality of T cells, TAMs can also promote apoptosis of T cells in GBM by expressing T-cell immunoglobulin and mucin domain-containing molecule 4 (TIM-4) [83,94], which is known to be a receptor of phosphatidylserine (PS) [94,95]. Interestingly, human glioma-derived T cells show higher expression of PS, which may boost T cell-lysis [94].

### 5.2. Spatial Distribution of Macrophages and Microglia in the TME 

TAMs are not evenly distributed throughout the tumor and rather concentrate around different anatomical structures. TAM-enriched areas often include hyperplastic blood vessels, peri-necrotic regions and areas of microvascular proliferation. Microglia are found near the edge of the tumor, while blood-derived macrophages are found near blood vessels and necrotic areas [34]. Hypoxic/necrotic areas are common in solid malignant tumors, including GBM. The distribution of TAMs with M1 and M2 phenotypes is dependent on the degree of hypoxia in the surroundings. The M2 phenotype is often found in the more hypoxic areas of the tumor. Experimental studies regarding the role of hypoxia on macrophage polarization showed interesting results [96,97]. Recently, hypoxic glioma-derived exosomes containing IL-6 and miR-155-3p were shown to activate the STAT3 pathway leading to autophagy which induced M2 polarization and promoted glioma progression [98]. In this regard, others showed that hypoxia itself induces the production of CCL2, CCL5, CXCL12, CSF-1 and VEGF that stimulates monocyte recruitment to the hypoxic areas [99]. When TAMs arrived at the TME, they increase immunosuppressive activity by sensing hypoxia-inducible factors [100]. 

### 5.3. TAMs and the Interferon Response in Glioma

Type I and Type II interferons are regarded as major contributors to an anti-tumor immune response in many cancers and were reviewed in detail elsewhere [101,102]. Interferon-β, a major type I interferon, is also known to have a direct growth-inhibitory effect on tumor cells, as demonstrated in GBM models in several studies [103,104]. Interferon-β impacts tumor cell proliferation and apoptosis. In addition, it can activate a pro-inflammatory program in different types of immune cells, including macrophages/microglia [101]. A recent study showed that another type-I interferon, interferon-α-14, was induced in macrophages upon niacin (Vitamin B3) exposure, which had a growth-inhibitory effect on glioblastoma cells [105]. Type I interferon genes are located on chromosome arm 9p, which is often deleted in gliomas and thus supports the hypothesis that these genes are tumor suppressors [106]. However, recent studies in experimental glioblastoma have pointed towards a more critical view of interferons. Studies by Maire et al. and Pollard et al. identified an interferon response signature in their respective GBM model systems, which contributes to immune evasion of tumor cells. Both studies also show that this signature is induced by interferons secreted by myeloid cells in the TME, such as macrophages [77,78]. These new data introduce more controversy regarding the role of interferons in glioblastoma and support the well-known functional diversity of cytokines in general. Another aspect to point out is the duration of interferon secretion in the TME. Chronic exposure to interferons, which is often observed in cancers, may indeed induce immune dysfunction [107]. Thus, anti-cancer treatment with cytokines should be regarded with more caution in the future, and chronic exposure should most likely be avoided.

### 5.4. Prognostic Relevance of TAMs in Glioma

It was proposed that TAM density and especially a switch towards an M2 polarization state correlate negatively with overall survival in glioma. This was also supported when analyzing samples from GBM patients receiving antiangiogenic treatment at recurrence [108], showing that higher numbers of CD11b^+^ cells correlated with a poor outcome. Furthermore, expression of the M2 marker CD168 by TAMs in IDH-mutated anaplastic astrocytoma was negatively correlated with survival [109]. Nevertheless, opposing results regarding prognostic markers are very common, which makes the prognostic relevance of TAMs uncertain. For instance, Zeiner et al. found that TAMs expressing high levels of CD68, CD163 and CD206 in the vital tumor core in GBM were associated with a better prognosis. Furthermore, mRNA expression profiling demonstrated that a mixed M1/M2 profile in the tumor core was associated with increased overall survival [36]. Another recent study observed that a higher portion of microglia in the TAM population significantly correlated with increased patient survival, indicating that monocyte-derived TAMs may have stronger tumor promoting properties compared to microglia [110]. Sørensen et al. used the M2 macrophage marker CD204 to identify tumor promoting macrophages in high-grade glioma and observed an unfavorable prognostic outcome [111]. In summary, these studies show conflicting results regarding the prognostic relevance of macrophages in glioma. The discrepancy may partially have emerged from macrophage/microglia heterogeneity. In addition, the use of different markers in these studies to identify macrophages may also have contributed to this discrepancy. More studies are needed to identify subpopulations of macrophages and microglia and their functional relevance for tumor growth and/or immunosuppression.

### 5.5. Experimental Models to Investigate the TME in Glioma

Due to the complexity of the TME with many stromal cells involved, in vivo models are, in general, a natural choice to investigate the TME in glioma. Several types of models are employed so far to study the TAM compartment in gliomas. In particular immunocompetent orthotopic models are used extensively because they include all cells of the TME that are also found in human GBM patients. Although both rat and mouse orthotopic models are widely used [112], mouse models have an advantage regarding immunology and immunotherapy research, as the immune system is better characterized in mice compared to rats. Consequently, more analytical tools and antibodies are available for mouse models. However, a major drawback, is the nature of glioma cell lines on which these models are based. The most widely used mouse cell line is GL261, followed by CT2A and others [113]. Both CT2A and GL261 were chemically induced in C57/BL6 mice a long time ago and were since employed as in vivo models. Apart from the mutational landscape, which is entirely different compared to human GBMs, both cell lines are virtually non-invasive, thus failing to replicate the invasive nature of human GBM, which is also a major therapeutic challenge. Although studies have claimed that both GL261 and CT2A are invasive [113,114], these cell lines do not show the profound single-cell invasion that is observed in human GBMs. In addition, the TME of these models does not completely replicate that of humans, although there are some similarities in particular regarding immunosuppressive macrophages [115]. Furthermore, GL261 responds to an immune checkpoint blockade [116], while in contrast, human GBMs show resistance to the same treatment [117]. Thus, the GL261 model may, in general, be more susceptible to immunotherapies, which is a discrepancy compared to human GBMs. Unfortunately, there is very little activity in the field towards generating better immunocompetent mouse/rat models for GBM. One encouraging study should be mentioned where Costa et al. developed glioma stem cell (GSC) lines based on GBM cells from genetically induced models [118]. These models are highly invasive and replicate the geno- and phenotype of human GBM much better compared to GL261 and CT2A. Such models could, in the future, bring basic and translational research closer to the human situation, which is urgently needed. Genetically engineered models could, in general, be a good alternative as they show invasive features and some of the genetic traits of human GBM; however, performing controlled therapeutic experiments is a challenge. Therefore, developing GSC lines based on these models, as shown by Costa et al., is an attractive option for the future [118]. Another attractive alternative to syngeneic models is humanized mouse models. However, these models are by far more expensive when compared to the syngeneic ones. In humanized mouse models, immune cells from different sources/organs such as peripheral blood, bone marrow, spleen or fetal liver are injected into severely immunodeficient mice. Recent activities in this field resulted in the establishment of many different humanized mouse models which were reviewed elsewhere [119]. A big advantage of these models is that human patient-derived GBM cells can be used, which are more clinically relevant compared to rodent tumor cells.

## 6. Therapeutic Options for Targeting TAMs

As TAMs contribute to glioma progression and play an important role in counteracting immunotherapies, there are potential strategies to target them. Indeed, various strategies to target the TAM population in GBM were identified and tested in experimental models and clinical settings. The TAM-targeting approaches can be conceptually divided into three groups (Figure 1). In TAM re-education, target proteins are inhibited by using monoclonal antibodies or small molecule inhibitors with the aim to re-polarize M2-like TAMs towards an M1-like phenotype. The second approach, designated herein as TAM education, involves the activation of pro-inflammatory circuits in TAMs. By using agonistic antibodies or other molecules, key proteins in the proinflammatory pathway can be activated. Such proteins can also be delivered by direct administration or gene therapy. Finally, the last group of TAM-targeting strategies involves depletion of TAMs. In this approach, unbiased depletion of TAMs or inhibition of macrophage infiltration is achieved by targeting key molecules. Individual points of potential intervention (as outlined in Table 1) are discussed below.

### 6.1. TAM Re-Education

TAM re-education describes a therapeutic approach where key target proteins that polarize TAMs towards an immunosuppressive M2 phenotype are inhibited. This inhibition will, in theory, lead to the M1 polarization of macrophages. This approach, if effective, may be one of the most promising since M1 macrophages have the potential to attract and activate adaptive immunity for effective tumor killing and a potential long-lasting anti-tumor immune response.

#### 6.1.1. Programmed Cell Death Protein 1 (PD-1)/PD-L1 Axis

PD-1 is a widely known immune checkpoint. Upon binding the cognate receptor PD-L1, often found on tumor cells and TAMs, PD-1 triggers reduced proliferation and activity of T lymphocytes. Thus, the PD-1/PD-L1 axis provides means for immune evasion [133] and very importantly, it impedes various immunotherapeutic interventions during cancer treatment. As a result, inhibitors of PD-1 have emerged as a promising group of cancer therapeutics. Indeed, anti-PD-1 treatment with monoclonal antibodies has indeed revolutionized the treatment paradigm for some cancer types such as metastatic melanoma and non-small cell lung carcinoma [134,135]. However, the situation appears bleak for GBM, where anti-PD-1 treatments failed to show efficacy in clinical trials [136,137]. Various reasons are attributed for this lack of efficacy in GBM, and immunological uniqueness (rather than immunological privilege) [138] of the brain is thought to be the root cause in this context. Spatial seclusion of the brain itself poses challenges for immunotherapies regardless of the tumor type. For example, peripheral metastases from renal cell carcinoma are almost two times more sensitive to anti-PD-1 therapy than brain metastases of the same tumor [139], and in a murine glioma model, orthotopic tumors are far more refractory to ICI than subcutaneous tumors [140]. Since T cells are the major effectors in anti-PD-1 treatment, T cell scarcity [73] and dysfunction [141] in GBM patients are likely to be a major reason behind this failure.

However, the absolute lymphocentric modus operandi of the anti-PD-1 treatment was challenged, as macrophage polarization towards an M1 phenotype was likely responsible for the therapeutic efficacy of an ICI regimen in a preclinical glioma model [56]. In line with this, a recent study by Rao et al. reported that the anti-PD-1 treatment can trigger an anti-tumor immune response even in the total absence of CD8^+^T cells in CD8^−/−^ GBM preclinical models [51]. Absence of CD8^+^T cells induced massive infiltration of TAMs, which upon anti-PD-1 treatment, demonstrated an M1 signature and killed tumor cells directly by antibody-dependent cell cytotoxicity mechanisms [51]. In addition, Rao et al. demonstrated that the anti-PD-1 monoclonal antibody was able to cross the BBB efficiently, where it reduced the number of PD-1^+^ microglia. Such amenability of TAMs in the context of anti-PD-1 treatment is not utterly surprising as PD-1 expression on TAMs is well-known, which causes poor phagocytic activity [142]. Since anti-PD-1 treatment can act through TAMs, the major immune cell population in GBM tissue, the prospect of this treatment regimen in GBM patients is still present [134] in spite of the failure of the phase III CheckMate 143 trial [137]. The result of the CheckMate 143 trial is also subject to cautious interpretation. In this large-scale trial, Nivolumab, the anti-PD-1 monoclonal antibody, was tested as a monotherapy. However, the anti-PD-1 single treatment was shown to only modestly increase survival in animal models where it required additional treatment(s) to provide pronounced therapeutic effect [56,143], and in agreement, a clinical trial with anti-PD-1 adjuvant settings showed that this therapy in combination with another treatment modality can be beneficial [120]. Thus, a key question is which combination treatment should be most attractive to approach further. Clinical trials for anti-PD-1 treatment in combination with radiotherapy (CheckMate 498, ClinicalTrials.gov Identifier: NCT02617589) and chemoradiotherapy (CheckMate548, ClinicalTrials.gov Identifier: NCT02667587) are currently ongoing where nivolumab is being tested in newly diagnosed GBM patients. Owing to the fact that adjuvant Nivolumab before debulking surgery showed promising results [120], oncolytic virotherapy or suicide gene therapy [56,144,145,146] in combination with anti-PD-1 treatment could be rational choices. Since PD-1-expression in the TAM compartment is heterogenous and not all TAMs are positive for PD-1 [51], a second TAM-targeted therapy may also be relevant in this case. One such example is the concomitant use of an Arginase inhibitor that increased the anti-tumor activity of a PD-1 inhibitor in murine gliomas [147]. However, the significance of Arginase and its inhibition in human GBM will have to be clarified before clinical translation of this strategy.

PD-L1, the cognate ligand of PD-1, is expressed on the surface of certain glioma cells and TAMs. Blocking PD-L1 can also destabilize the PD-1 mediated immunosuppressive activities, and thus, targeting PD-L1 has also emerged as a promising option for a TAM-targeting glioma treatment strategy. Monoclonal antibodies targeting PD-L1 are safe for patients [148]. However, resembling the anti-PD-1 treatment, anti-PD-L1 has a superior effect in non-CNS solid cancers compared to GBM [149,150], and thus this strategy will most likely require a combinatory regimen to increase its efficacy for GBM patients.

#### 6.1.2. Cytotoxic T-Lymphocyte-Associated Protein 4 (CTLA4)

CTLA4 (also known as CD152) is one of the most studied immune checkpoints and a major negative regulator of T-cell activation. Monoclonal antibodies against CTLA4 represent one of the earliest ICI treatment regimens developed and are tested in various cancers, including GBM [138]. Currently, several anti-CTLA4 clinical trials are ongoing for GBM treatment in different combinations (ClinicalTrials.gov Identifier: NCT03233152 and NCT02017717). While anti-CTLA4 treatment is a promising therapeutic option for GBM, the significance of TAMs in this process is not clear. Saha et al. reported the involvement of both resident microglia and infiltrating macrophages in the therapeutic efficacy of a double ICI regimen (anti-CTLA4 and anti-PD-1) in combination with oncolytic immunovirotherapy [56]. However, a single anti-CTLA4 treatment was not tested in this context, and thus it is not clear if the CTLA4-mediated efficacy involves TAMs. Expression of CTLA4 was reported on dendritic cells [151], but it is not well-documented if TAMs express CTLA4. Given that ICI monotherapy may not be successful for GBM designing a rational combinatory regimen in conjunction with CTLA4 may also be a necessary step. In this context, it is very important to unravel any potential significance of TAMs for the anti-CTLA4-mediated immunotherapeutic regimen.

#### 6.1.3. CD47

CD47, a broadly expressed membrane protein, functions as a “don’t eat me” signal through binding to its corresponding receptor signal regulatory protein-α (SIRPα), mostly expressed on granulocytes and macrophages, including TAMs [152]. CD47 is expressed on all normal cells and functions as a safety mechanism to prevent phagocytosis. However, CD47 is frequently upregulated on cancer cells compared to normal cells and thus is regarded as an attractive target for cancer immunotherapy [153]. Inhibiting CD47 hinders cancer immune escape and increases phagocytic activity. Inhibition of CD47 also down-regulates M2 macrophage markers and thereby strengthens the pro-inflammatory response [154]. Hematopoietic cancers, in particular lymphoma, were identified as attractive targets for treatment with anti-CD47 or anti- SIRPα antibodies [155]. A recent clinical trial for Non-Hodgkin’s Lymphoma showed very promising results using CD47 antibodies in combination with rituximab [156]. Data on solid cancers also showed some efficacy of anti-CD47 targeting strategies in experimental models [157,158]. However, CD47 is not the only immune checkpoint for macrophages on solid cancer cells [159,160] and inhibiting CD47 alone might not be efficient enough to achieve a substantial treatment effect. Several experimental studies using anti-CD47 antibodies were also performed in GBM with contrasting results [154,161,162]. To conclude, as with most solid cancers, the treatment with CD47 antibodies alone is not sufficient, but the therapeutic effect may increase in combination with other treatment modalities [161,163]. The treatment with anti-CD47 antibodies in GBM seems to affect both microglia and TAMs from the periphery, suggesting that CD47 inhibition may also re-educate the resident microglia population [121,163].

#### 6.1.4. CD73

CD73, also known as ecto-5′-nucleotidase, is an important ectoenzyme in the purinergic signaling pathway that involves the conversion of ATP to adenosine. CD73 expression is found in various tumor cells, including GBM [164] and immune cells of both myeloid and lymphoid lineages [165], with variable frequencies depending on homeostatic or diseased cellular states [166]. Unlike ATP, a damage-associated molecular pattern molecule, adenosine, functions as an immunosuppressive molecule. Upon release from the stressed cells or by extracellular catalysis of adenine nucleotides, adenosine regulates tissue remodeling activities which further involves recruitment of regulatory T cells and pro-tumorigenic CD11b^+^ cells [165]. In concert with this function, CD73 is shown to promote the growth of GBM, and high CD73 expression in GBM patients represents a negative prognostic factor [46,164]. In contrast, suppression of CD73 can result in reduced GBM growth. Goswami et al. showed that GBM tissues harbor a unique cluster of TAMs with high CD73 expression, and it persists after ICI treatment [46]. This CD73^high^ TAM population demonstrated an immunosuppressive expression signature, and not surprisingly, an M1-polarized immune microenvironment was observed in CD73-KO murine glioma models [46]. The absence of CD73 also significantly increased the effect of ICI treatment in this murine model.

Consequently, a number of anti-CD73 treatment strategies emerged over the last few years. Azambuja et al. developed a siRNA-based anti-CD73 approach that can be delivered intranasally and showed that this treatment can reduce tumor growth in rat glioma models [164]. However, this strategy is yet to be tested in a clinical setting. A group of monoclonal antibodies (e.g., CPI-006, Sym024, INCA00186 and NZV930) and small-molecule-inhibitors (e.g., Quemliclustat and LY3475070) were also developed recently, and some of these are being tested clinically for various non-CNS cancer types [167]; however, not yet for GBM. The preclinical success of anti-CD73 combined with ICI-treatment in CD73-KO models warrants further research in this direction. In the future, it would be important to test the bioavailability, functionality and toxicity of these inhibitory molecules for a meaningful clinical translation for GBM treatment.

#### 6.1.5. STAT3

Signal transducers and activators of transcription 3 (STAT3) is a transcription factor that is important for tumor development in various cancers, including GBM. It is activated by direct phosphorylation of receptor tyrosine kinases as well as Janus kinases (JAKs) [168]. In addition to pro-tumorigenic stimuli, STAT3 activation induces immunosuppression in the TME due to reduced expression and secretion of pro-inflammatory cytokines such as IL-12 and TNF-α by tumor cells [168]. Inhibition of STAT3 hinders tumor growth and induces apoptosis in gliomas in addition to suppression of TAM-activation. STAT3 also alters the immune cells in the TME by favoring enrichment of immunosuppressive regulatory T-cells and increasing M2 polarization [122]. In vitro, the inhibition of STAT3 suppresses M2 polarization and tumor cell proliferation in human GBM cells [169]. When macrophages and T-cells from GBM patients were treated with a STAT3 inhibitor in vitro, macrophages increased production of IL-12 and induced a Th1 shift leading to T cell activation [122,170]. Thus, STAT3 represents a potential target in combinatorial immunotherapy by stimulating T-cell activation. 

#### 6.1.6. Macrophage Receptor with Collagenous Structure (MARCO)

MARCO is a class A scavenger receptor expressed by TAMs. High expression is shown to drive GBM towards a mesenchymal phenotype and is correlated with accelerated tumor growth and a poor clinical outcome [171]. Inhibitory anti-MARCO monoclonal antibodies showed promising results in preclinical mouse models of melanoma, breast cancer and colon cancer. An anti-tumor effect was observed for both breast cancer and melanoma and resulted in a TAM shift towards a more pro-inflammatory phenotype. The expression of MARCO was mainly found in M2 polarized TAMs and increased when stimulated with IL-10 and TGF-β stimulation. Interestingly, the anti-MARCO treatment showed higher efficacy when combined with anti-CTLA4 immune checkpoint inhibitors [123]. 

#### 6.1.7. CXCL16/CXCR6 Axis

The transmembrane chemokine CXCL16 and its corresponding receptor CXCR6 play an important role in neuroprotection during ischemic insults of the brain [172]. CXCL16 is highly expressed in human gliomas compared to a normal brain, where it is mainly found in vascular endothelial cells. Furthermore, CXCR6 is found in astrocytes and microglia [173]. Lepore et al. observed that CXCL16 promotes an immunosuppressive milieu by interaction with CXCR6 expressed on TAMs. In vitro experiments showed microglia polarization towards M2 when treated with CXCL16. Moreover, glioma bearing CXCR16-K.O. mice survived longer compared to wild type mice, and CXCL16/CXCR6 stimulation promoted tumor progression in human GBM [124]. Inhibition of CXCL16/CXCR6 is not yet tested in clinical trials for glioma.

#### 6.1.8. WISP1

The Wnt/β-catenin signaling pathway, which is highly activated in GBM, regulates cell proliferation, migration and cell death. High expression of WISP1 is associated with poor prognosis in most cancers, including GBM [174,175]. Tao et al. observed that WISP1 is the only highly expressed wnt/β-catenin target gene in GBM compared to normal brain tissue. The authors showed that WISP1 knockdown in glioma xenografts lead to a decreased TAM density. When analyzing the distribution of M1/M2 TAMs, WISP1 knockdown markedly reduced M2 macrophages but had little effect on M1 macrophages. This indicates that WISP1 promotes an immunosuppressive milieu. It was also observed that WISP1 and its integrin receptor α6β1 were expressed at higher levels in M2 macrophages relative to M1 macrophages [125]. Treatment with carnosic acid, a molecule that inhibits β-catenin activity, showed increased survival in experimental GBM [125]. The effect of carnosic acid was increased by combination with Temozolomide, which intensified cell-cycle arrest and increased apoptosis [176]. Yet, clinical trials using carnosic acid for the treatment of GBM were not initiated. 

### 6.2. TAM Education

TAM education includes therapeutic approaches where proinflammatory reactions are induced in TAMs by agonistic antibodies, small molecules or other means. Similar to TAM re-education, this approach is quite promising as the aim is to induce a phenotypic switch from M2 to M1 macrophages.

#### 6.2.1. CD40/CD40L

The CD40 transmembrane receptor and its ligand CD40L (CD154) belong to the TNF/TNFR superfamily [126]. CD40 functions as a co-stimulatory molecule and is expressed in a variety of immune cells, including macrophages, dendritic cells and B cells [127]. CD40 is also expressed in the resident microglia population, where it is upregulated during microglial activation and plays a decisive role in autoimmune inflammation [177]. Upon binding its cognate ligand CD40L, which is either secreted by or expressed on the surface of activated CD4^+^ T cells [178], CD40 increases the antigen-presenting function of macrophages and orchestrates an anti-tumor immune response by leading to the secretion of a variety of pro-inflammatory molecules [126]. As a result, the CD40/CD40L axis offers prospects for exploitation, either by providing exogenous CD40L or targeting CD40 with agonistic antibodies. Direct administration of recombinant CD40L was pursued in non-CNS tumors with a promising safety profile [179] and is being tested as a fusion protein with SIRPα to initiate checkpoint blockade (ClinicalTrials.gov Identifier: NCT04406623). Gene therapy technology is also being harnessed to deliver CD40L to tumor tissue using engineered Vaccinia virus [180] and adenoviral vectors [181] to treat non-CNS tumors. In the context of glioma, the administration of agonistic antibodies against CD40 was tested in preclinical models [127,182]. Using the anti-CD40 monoclonal antibody FGK4.5 [61] in combination with celecoxib, a COX-2 inhibitor, Kosaka et al. reported significant survival benefits in the murine Gl261 tumor model [182]. This combination therapy triggered the expression of CXCL10 and downregulation of Arginase 1 in the CD11b^+^ cells, indicating an M1-like polarization of the myeloid cells. Depletion of T cells in this model abolished the therapeutic effect, which is in line with the antigen-presenting functionality of the CD40/CD40L axis. Administration of FGK4.5 in combination with irradiated tumor cells also increased survival in glioma murine models [127]. Interestingly, CD40, and also CD40L to some extent, were found to be expressed by human gliomas and the CD40/40L co-expression in glioma patients serves as a positive prognostic factor [127]. One interesting and remaining question in the context of FGK4.5 treatment is if anti-tumor immunity is generated only extracranially or if sufficient antibody bioavailability can be achieved in the tumor bed to activate resident microglia as well. Even though CD40 antagonistic antibody treatment shows auspicious results in glioma preclinical models, it has not yet been tested in clinical trials for GBM treatment.

#### 6.2.2. Toll-like Receptor (TLR) Activation

Humans have at least 10 TLR homologues of Toll—an important protein involved in the insect immune defense [183]. TLRs are membrane glycoproteins and belong to the interleukin-1 receptors superfamily. TLRs expression is widely found in various normal cells, tumor cells including gliomas and innate immune cells such as macrophages and dendritic cells [184]. While the expression of TLRs in tumor cells may have immune system-independent functions, TLRs expressed on macrophages and dendritic cells orchestrate potent immune reactions after recognizing a variety of pathogen-associated molecular patterns (PAMPs). PAMP-binding initiates an active innate immune response characterized by secretion of various pro-inflammatory molecules and increased phagocytosis [128]. Thus, it is not surprising that TLR-triggering has emerged as an attractive strategy for cancer immunotherapy [128,184]. Various TLR agonists such as Monophosphoryl lipid A for TLR4, Polyinosinic:polycytidylic acid (abbreviated poly I:C) for TLR3, CpG DNAs for TLR9 are being developed for the immunotherapy of various solid cancers [128,184]. Targeting TLR3 (originally recognizes viral dsRNA) and TLR9 (originally recognizes unmethylated CpG DNA from pathogens) showed some efficacy in experimental GBM models [185]. Indeed, several clinical trials were previously conducted with agonists for TLR3 or TLR9 for the treatment of glioma, albeit with limited success [128,129,186] (ClinicalTrials.gov Identifier: NCT02149225). As TLR3 and TLR9 follow two different pathways, MYD88-independent and -dependent, respectively, a combination of both was suggested that could yield a synergistic effect [185]. However, the issue of poor T cell response in GBM [73,141] is a huge concern that can likely explain the limited success of TLR-agonists in glioma clinical trials. Thus, a combination of TLR agonists with other immunotherapies such as ICIs may be a rational choice for further testing [187]. Currently, clinical trials using TLR3 agonists in combination with GM-CSF (ClinicalTrials.gov Identifier: NCT02149225) or peptide vaccine (ClinicalTrials.gov Identifier: NCT01920191) for treatment of GBM are being initiated.

### 6.3. TAM Depletion

TAM depletion aims at substantially depleting the immunosuppressive TAM population by targeting proteins that promote macrophage survival, attraction or entrance into the TME. However, this approach may not be a straightforward one as macrophages/microglia might be needed in the context of immunotherapies to contribute to an anti-tumor immune response.

#### 6.3.1. P-Selectin

P-selectin (SELP) is an important adhesive molecule in leukocyte rolling and is shown to have promising therapeutic implications. SELP is expressed by different stromal cells, including tumor cells and TAMs [188]. SELP-corresponding ligand P-selectin glycoprotein ligand-1 (PSGL-1) is expressed on T-cells and TAMs. Moreover, PSGL-1 is regarded as an immune checkpoint, directly impacting the immune response [189]. Yeini et al. showed that co-culturing glioma cells and microglia led to an enhanced secretion and expression of SELP when compared to monoculture. SELP knockdown in glioma cells reduced proliferation and migration when co-cultured with microglia compared to controls. In addition, the authors found that high SELP expression in GBM was correlated with short term survival (<2 months) and not with long term survival (>5 years). They also observed that the SELP-PSGL-1 axis drove microglia towards a more immunosuppressive phenotype with increased expression of ARG-1, IL-10 and TGB-β in animal studies [130]. Despite these promising preclinical results, and although P-selectin monoclonal antibody is accepted by FDA for use in sickle cell anemia to prevent pain crisis [190], clinical trials are not yet being initiated for glioma treatment.

#### 6.3.2. CCL2/CCR2

CCL2 promotes the efflux of CCR2^+^ monocytes into the tumor bed [18,191]. CCR2 also binds CCL7, CCL8 and CCL12, but apart from CCL2, only CCL7 was found to be able to induce monocyte infiltration [18]. Total TAM number may be associated with aggressiveness and poorer prognosis in GBM [111], and CCL2 expression promotes glioma growth in animals models [192] in addition to being a negative prognostic factor in GBM [193]. Thus, the CCL2/CCR2 axis appears to be a potential point for therapeutic intervention. Indeed, Zhu et al. showed that glioma-bearing CCL2^+/−^ heterozygous mice or a neutralizing antibody treatment for CCL2 in both human and murine glioma models increased survival [131]. The survival was nevertheless modest, possibly accounting for partial blocking of CCL2, while the addition of TMZ treatment increased the therapeutic effect further. Although the therapeutic efficacy of CCL2/CCR2 inhibition is well-documented in preclinical GBM models, there are few concerns regarding this strategy. Firstly, CCL2/CCR2 inhibition does not target the resident microglia population which are normally devoid of CCR2 [193]. CCL2 inhibition targets only the Ly6C^high^ monocytes, whereas the Ly6C^low^ monocytes have poor CCR2 expression and thus may escape this treatment [18] and still modulate pro-tumorigenic functions. Secondly, this type of unbiased macrophage/monocyte depletion may rather inhibit than support anti-cancer immunity. TAMs are important antigen-presenting cells in the GBM microenvironment and thus play a key role in mounting the adaptive immune system. Severe, unbiased macrophage/monocyte depletion may therefore hinder an immunotherapeutic regimen [56] and activation of adaptive immunity, which is counterproductive from a therapeutic perspective.

#### 6.3.3. Colony Stimulating Factor 1 Receptor (CSF1R)

CSF1R, which is also known as macrophage colony-stimulating factor receptor or CD115, controls the differentiation and function of macrophages and microglia. In normal conditions, CSF1R signaling is indispensable for the survival of these cells. The anti-tumorigenic potential of CSF1R inhibition is documented in various cancers [194,195], and thus a variety of anti-CSF1R molecules (monoclonal antibodies and small-molecule inhibitors) were developed [196]. Nevertheless, a clinical trial using a CSF1R inhibitor in recurrent glioma patients did not increase survival [197]. This failure may be explained by two key issues. Firstly, the efficacy and effect of CSFR1 inhibition is highly dependent on tissue type and disease condition and can either lead to depletion of certain macrophage populations or repolarization of existing macrophages. In the context of allogeneic hematopoietic cell transplants, for example, where macrophages play an immunosuppressive role to limit the acute graft-versus-host disease (GvHD), anti-CSF1R treatment inhibits the total macrophage population and thereby aggravates GvHD [198]. Similarly, in the healthy adult brain, CSF1R inhibitors deplete microglia cells very efficiently (up to 99% or more), and drug withdrawal permits rapid repopulation of microglia cells in a CSF1R-dependent manner [199]. However, for TAMs, the situation appears to be different. PDGF-driven GBMs, for example, can support TAMs during CSF1R-inhibition and protect from depletion [132]. Cytokines secreted by GBM cells, such as GM-CSF, IFN-γ and CXCL10, are associated with this process. Interestingly, even though CSF1R-inhibition fails to deplete TAMs in this condition, profound phenotypic changes occur whereby the TAMs demonstrate an M1 expression signature and contribute to increased survival in murine PDGF-driven glioma models [132]. Interestingly and importantly, this effect is not observed in all types of gliomas. In Gl261 murine glioma, which is also sensitive to anti-CSF1R treatment, CSF1R-inhibition leads to TAM depletion with a more pronounced effect on the microglial fraction compared to macrophages [200]. This result resembles the effect of CSF1R-inhibition in mammary- and hepatocellular carcinomas, where the anti-tumor effect is preceded by the depletion of TAMs [194,195]. Furthermore, in another murine glioma model, known as 005GSC, CSF1R-inhibition resulted only in modest depletion of TAMs and reduced therapeutic efficacy of an ICI regimen in combination with oncolytic immunovirotherapy [56]. In light of these divergent effects of CSF1R-inhibition in different glioma models and therapeutic settings, it will be important in the future to define mechanisms behind tumor-type-specific actions of CSF1R-inhibition.

The second explanation behind the failure involves the emergence of resistance towards the anti-CSF1R treatment, which for example, can be mediated by the production of insulin-like growth factor 1 (IGF1) by TAMs [201]. In line with this, the dual inhibition of IGF1R and CSF1R is sufficient to increase treatment efficacy further, and thus this combinatorial regimen may serve as a potential therapeutic option for clinical testing.

Currently, CSF1R-inhibition, in combination with chemoradiotherapy (ClinicalTrials.gov Identifier: NCT01790503) or with ICI (ClinicalTrials.gov Identifier: NCT02829723), is being tested for GBM in the clinical setting. Radiotherapy was previously shown to increase the therapeutic effect of CSF1R-inhibition in a PDGF-driven glioma model [202] and thus justifies clinical translation.

## 7. Conclusions

In this review, we have described how TAMs serve as a driving force for a pro-tumorigenic and immunosuppressive microenvironment found in malignant brain tumors. The TAM population in GBM is highly heterogeneous, and plasticity in their phenotype is well-documented. Another aspect of the varying and, to some extent, contradictory results in TAM research is the implications of different markers used to distinguish different TAM populations. Despite the availability of a set of well-accepted markers, a fluctuating expression depending on the TME makes it difficult to interpret the complex in vivo findings. As an example, microglia typically have low CD45 expression, but in the context of glioma, they may upregulate CD45 and can be mistaken for being macrophages of hematopoietic origin [66]. Thus, the choice of markers may introduce a potential bias when describing the contribution of different types of TAMs in the TME. While it is more or less widely accepted that most TAMs are in an M2 state, this might be due to the restricted use of surface markers such as CD163 and CD204 together with cytokines such as TGF-β and IL-10 to classify M2. Keeping in mind that TAMs can express both M1 and M2 polarization markers, many of these descriptions might not reveal the complete picture [203]. The difficulties in finding a specific marker for TAM origin and phenotype affirm its considerable plasticity and that the simplistic view of an M1/M2 division is by far an oversimplification. 

Since TAMs are important players in treatment resistance, they also offer potential options for targeted therapies for glioma treatment. Indeed, a variety of potential targets were identified and tested in both preclinical and clinical settings. Various strategies are used as means of such interventions—antagonistic or agonistic antibodies, small molecule inhibitors, synthetic molecules, nucleic acids, gene- and virotherapy, etc. Although some of these strategies have resulted in better therapeutic outcomes in preclinical models of glioma, no single agent has so far succeeded in clinical trials. Considering the complexity and cross-talk between stromal and tumor cells in GBM, identifying one single drug which is effective seems to be unrealistic at this point. It is more rational to think that a combinatorial approach will result in a better outcome. In conclusion, more research regarding combinations of promising immunotherapies is highly demanded for the future treatment of GBM patients. 

## Figures and Tables

**Figure 1 cancers-14-01319-f001:**
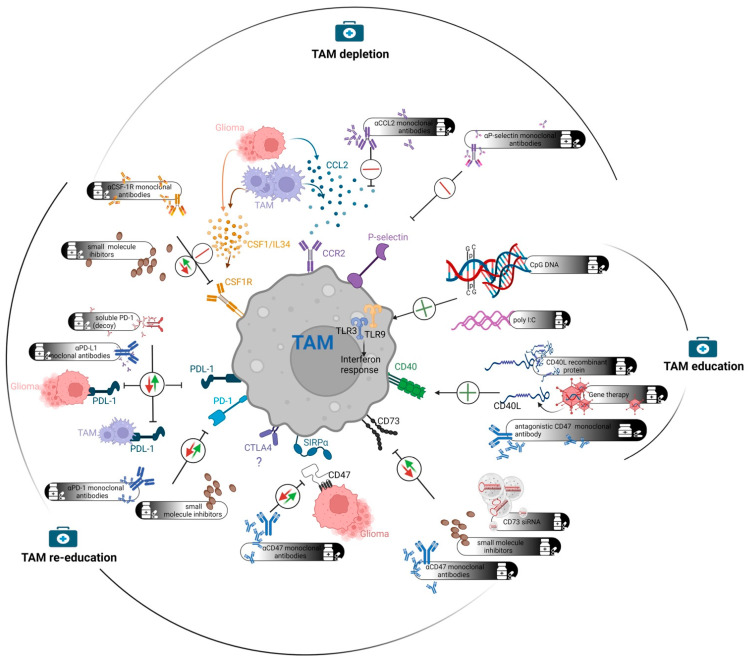
Targeting TAMs in GBM. Strategies of targeting TAMs can be categorized into three distinct approaches. TAM re-education involves targeting the immune checkpoints that provoke an immunosuppressive M2 phenotype. ICI regimens belong to this category. Depending on the glioma model used, CSF1R-inhibition demonstrates either TAM-reprogramming or depleting capabilities. TAM education activates the pro-inflammatory machineries of myeloid cells that can promote phagocytic activity, antigen presentation and the secretion of an array of immune-activating cytokines. TAM depletion targets the ligands and receptors that are involved in macrophage/monocyte infiltration from the periphery. These approaches are not mutually exclusive in a clinical setting. Particular combinations, which need to be tested more thoroughly in the future, could potentially lead to a better outcome.

**Table 1 cancers-14-01319-t001:** Potential options for immunotherapy in GBM.

Target	Mechanism	Effect of Treatment
TAM Re-education		
PD-1/PD-L1	PD-L1 binding to PD-1 receptors serves as inhibitory signaling for T cell activation and TAMs [120].	Inhibition: Increased T-cell proliferation, activity and killing by both T cells and TAMs.
CTLA4	Negative regulator of T-cell activation [56].	Inhibition: Increased T-cell activation. Uncertain effects on TAMs.
CD47	CD47 binding to SIRPα on TAMs serves as a “don’t eat me” signal [121].	Inhibition: Increased phagocytic activity. Reduced M2 markers.
CD73	Converts ATP to adenosine which stimulates immunosuppression [46].	Inhibition: Reduced tumor growth and increased M1 polarization.
STAT3	STAT3 activation stimulates tumor growth and reduces TAM activation [122].	Inhibition: Decreased tumor progression and increased immune response.
MARCO	MARCO stimulation drives GBM towards mesenchymal shift. Induce immunosuppressive TAMs [123].	Inhibition: Increased pro-inflammatory TAMs. Increased effect of CTLA2 in melanoma and breast cancer.
CXCL16	CXCL16 binding to its receptor CXCR6 promotes tumor progression and M2 polarization in microglia [124].	Inhibition: Increased M1 markers. Reduced tumor growth.
WISP1	WISP1 binding to α6β1-receptor induces immunosuppression and tumor progression [125].	Inhibition: Reduced immunosuppression. Reduced tumor growth.
TAM Education		
CD40/CD40L	Increased antigen-presenting function and immune response [126,127].	Activation: M1-like polarization and increased survival.
Toll-like receptor	Activates innate immune response [128,129].	Activation: Limited success in the treatment of glioma.
TAM Depletion		
SELP	SELP-PSGL-1 axis increases M2 shift [130].	Inhibition: Reduced immunosuppression. Reduced tumor growth.
CCL2/CCR2	Monocyte chemoattractant and promotes efflux of monocytes [131].	Inhibition: Reduced influx of infiltrating TAMs. Increased survival in animal models.
CSF1R *	CSF1R stimulation induces differentiation and survival of TAMs [132].	Inhibition: Reduced immunosuppression. Reduced tumor growth.

* Depending on the model used, CSF1R-inhibition can be placed in either TAM Depletion or TAM re-education category.

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
