# Peer review of "Tumor-Associated Macrophages in Gliomas—Basic Insights and Treatment Opportunities"

_cancers, 2022, doi:10.3390/cancers14051319_

Round 1

Reviewer 1 Report

Reviewer comments and suggestions

The present review paper has studied tumor-associated macrophages specifically for glioblastoma (GBM). Tumor-associated macrophages are macrophages that make up a large component of tumor tissue in many cancer types. These macrophages are found in great numbers in gliomas, a kind of primary brain tumor. Tumor-associated macrophages have been shown to promote glioma growth and influence the success of different treatment options. The authors summarize the results of TAM as potentially targeted therapeutic options that involve these compartments in gliomas.  

Decision: Minor revision is needed.

The paper has nicely complied and covered utmost part related to GBM associated macrophages. However, in many places, the authors need to do a few small corrections in the manuscript. Based on my view, below are the comments that need to be incorporated in the revised version of the manuscript. 

  1. Line 32 CNS full form needed in the abstract
  2. Line no 51, ‘a’ should be small of Astrocytoma.
  3. Line 74-75 (“In addition to macrophages originating from bone marrow-derived monocytes, certain tissues host resident macrophages”) is not understandable. Please rephrase the sentence. 
  4. Line 94 (These so-called “find me”) use appropriate verb.
  5. Line 131-133, after citation 32 sentences might be connected with connecting words (and/while/however).
  6. Line 136, you have discussed CD14 and F4/80 in line 137-138 but not about CD68 and Iba. Write at least one important point about both.
  7. Line 338, ‘Costa et al.’ cite again in the text.
  8. Line 488, add space between CD73 and high.

    Reviewer comments and suggestions

    The present review paper has studied tumor-associated macrophages specifically for glioblastoma (GBM). Tumor-associated macrophages are macrophages that make up a large component of tumor tissue in many cancer types. These macrophages are found in great numbers in gliomas, a kind of primary brain tumor. Tumor-associated macrophages have been shown to promote glioma growth and influence the success of different treatment options. The authors summarize the results of TAM as potentially targeted therapeutic options that involve these compartments in gliomas.  

    Decision: Minor revision is needed.

    The paper has nicely complied and covered utmost part related to GBM associated macrophages. However, in many places, the authors need to do a few small corrections in the manuscript. Based on my view, below are the comments that need to be incorporated in the revised version of the manuscript. 

    1. Line 32 CNS full form needed in the abstract
    2. Line no 51, ‘a’ should be small of Astrocytoma.
    3. Line 74-75 (“In addition to macrophages originating from bone marrow-derived monocytes, certain tissues host resident macrophages”) is not understandable. Please rephrase the sentence. 
    4. Line 94 (These so-called “find me”) use appropriate verb.
    5. Line 131-133, after citation 32 sentences might be connected with connecting words (and/while/however).
    6. Line 136, you have discussed CD14 and F4/80 in line 137-138 but not about CD68 and Iba. Write at least one important point about both.
    7. Line 338, ‘Costa et al.’ cite again in the text.
    8. Line 488, add space between CD73 and high.

Author Response

We thank this reviewer for the comments.

Response to the individual comments:

  1. Revised
  2. Revised
  3. Revised
  4. Line 99 (previously line 94) < These so called “find me” signals guide these cells toward the apoptotic cells and enable binding to the “eat me signals”. > We think an appropriate verb has already been used.
  5. Revised
  6. Good point. We have included a few introductory sentences.
  7. Revised
  8. <superscript> applied

Reviewer 2 Report

In this review the authors focus on tumor associated macrophages and microglia in glioblastoma. Specifically, they highlight on the difference between microglia and macrophage in GBM and therapeutic options targeting TAM. The author should address the following question to improve the manuscript.

  1. Include more figures describing the difference between macrophage and microglia in GBM microenvironment.
  2. The authors should include a paragraph regarding how macrophages regulate T cells and NK activity in GBM tumor microenvironment.
  3. More recent references about the progress in the research area should be included in the review.

Author Response

We thank this reviewer for the comments.

Response to the comments:

  • While a number of specific biomarkers have been identified recently to distinguish microglia from bone marrow-derived macrophages, a complete picture of exclusive functional differences between these two types of cells is still lacking in current literature. Therefore, the authors think that a figure about the differences of microglia and macrophages may not be sufficiently informative and appealing within the scope of this review.
  • We thank this reviewer for putting forward the important topic of ‘TAM-lymphocyte’ interplay in GBM. We have included a new paragraph (under section 5.1 The Function of TAMs in the TME) describing how TAMs obstruct T cell immunity in GBM according to the reviewer’s comments. Interaction with NK cells is also another important aspect of TAM-biology. However, their crosstalk is much less explored in GBM. Furthermore, prominent TAM-targeting strategies, in principle, largely do not involve TAM-NK collaboration. Therefore, the authors think that the discussion on TAM-NK collaboration is dispensable in this manuscript.
  • The authors revisited the references of this manuscript and found that nearly 55% of the references in the previous version was from last 5 years. However, many important reports within the basic and translational macrophage/microglia/TAM biology field are older than 5 years. As we aimed to use the original sources for our citations, it was not possible to increase the ratio of more ‘recent’ citations. We have, nevertheless, searched again and included several very recent citations (2020 or onward) in the revised manuscript.

Reviewer 3 Report

The author systematically introduced and summarized the important role of TAM in Gliomas and discussed how to target TAM in tumor treatment. The manuscript is well written and easy to understand and follow. Below are two minor suggestions.

  1. It is better to combine part3 and part4 into one part. In part 3 and part 4. The author generally summarized the characteristic of Macrophage plasticity and TAM marker, but the author didn't talk too much about the microglia in this part. So it is better to combine these two parts together and the title could be like " TAM plasticity and marker".
  2. In the "Therapeutic options for targeting TAM" part, the author summarized how to target TAM in the clinic from three aspects (TAM re-education, TAM education, and TAM depletion). It is better to add a brief introduction at the beginning of each aspect to make the reader easy to understand and follow.

Author Response

We thank this reviewer for both comments.

Response to the comments:

1. We have further discussed the polarization issue of microglia as the reviewer pointed out. We also think that the ‘marker’ section deserves to be discussed separately and thus should not be merged with the preceding section.

2. We have revised the manuscript accordingly.

Round 2

Reviewer 2 Report

N/A